# Opioid-Sparing Analgesia Impacts the Perioperative Anesthetic Management in Major Abdominal Surgery

**DOI:** 10.3390/medicina58040487

**Published:** 2022-03-28

**Authors:** Miruna Jipa, Sebastian Isac, Artsiom Klimko, Mihail Simion-Cotorogea, Cristina Martac, Cristian Cobilinschi, Gabriela Droc

**Affiliations:** 1Department of Anesthesiology and Intensive Care I, ‘Fundeni’ Clinical Institute, 022328 Bucharest, Romania; miruna.jipa@stud.umfcd.ro (M.J.); mihailcotorogea@gmail.com (M.S.-C.); christtina_martac@yahoo.com (C.M.); gabriela.droc@umfcd.ro (G.D.); 2Department of Physiology, Faculty of Medicine, Carol Davila University of Medicine and Pharmacy, 020021 Bucharest, Romania; 3Laboratory of Molecular Neuro-Oncology, Department of Neurology, University Hospital Zurich, 8091 Zürich, Switzerland; artsiom.klimko@usz.ch; 4Department of Anesthesiology and Intensive Care, Clinical Emergency Hospital, 014461 Bucharest, Romania; cristian.cobilinschi@drd.umfcd.ro; 5Department of Anesthesiology and Intensive Care II, Faculty of Medicine, University of Medicine and Pharmacy, Carol Davila, 020021 Bucharest, Romania; 6Department of Anesthesiology and Intensive Care I, Faculty of Medicine, University of Medicine and Pharmacy, Carol Davila, 020021 Bucharest, Romania

**Keywords:** opioid-sparing anesthesia, ketamine, systemic lidocaine, epidural analgesia, major abdominal surgery

## Abstract

*Background and Objectives*: The management of acute postoperative pain (APP) following major abdominal surgery implies various analgetic strategies. Opioids lie at the core of every analgesia protocol, despite their side effect profile. To limit patients’ exposure to opioids, considerable effort has been made to define new opioid-sparing anesthesia techniques relying on multimodal analgesia. Our study aims to investigate the role of adjuvant multimodal analgesic agents, such as ketamine, lidocaine, and epidural analgesia in perioperative pain control, the incidence of postoperative cognitive dysfunction (POCD), and the incidence of postoperative nausea and vomiting (PONV) after major abdominal surgery. *Materials and Methods*: This is a clinical, observational, randomized, monocentric study, in which 80 patients were enrolled and divided into three groups: Standard group, C (*n* = 32), where patients received perioperative opioids combined with a fixed regimen of metamizole/acetaminophen for pain control; co-analgetic group, Co-A (*n* = 26), where, in addition to standard therapy, patients received perioperative systemic ketamine and lidocaine; and the epidural group, EA (*n* = 22), which included patients that received standard perioperative analgetic therapy combined with epidural analgesia. We considered the primary outcome, the postoperative pain intensity, assessed by the visual analogue scale (VAS) at 1 h, 6 h, and 12 h postoperatively. The secondary outcomes were the total intraoperative fentanyl dose, total postoperative morphine dose, maximal intraoperative sevoflurane concentration, confusion assessment method for intensive care units score (CAM-ICU) at 1 h, 6 h, and 12 h postoperatively, and the postoperative dose of ondansetron as a marker for postoperative nausea and vomiting (PONV) severity. *Results*: We observed a significant decrease in VAS score, as the primary outcome, for both multimodal analgesic regimens, as compared to the control. Moreover, the intraoperative fentanyl and postoperative morphine doses were, consequently, reduced. The maximal sevoflurane concentration and POCD were reduced by EA. No differences were observed between groups concerning PONV severity. *Conclusions*: Multimodal analgesia concepts should be individualized based on the patient’s needs and consent. Efforts should be made to develop strategies that can aid in the reduction of opioid use in a perioperative setting and improve the standard of care.

## 1. Introduction

According to the latest available World Health Organization global survey on surgery volume from 2014, at least 312 million surgeries are performed globally every year [1]. Major abdominal surgery accounts for a substantial part, involving emergency and elective procedures, as well as oncologic and non-oncologic surgery [2,3]. Some of those procedures (i.e., hepatic or pancreas surgery) are performed in reference centers for abdominal surgery [3,4,5].

Among these patients, up to 10% suffer from severe chronic postoperative pain that persists 6 months after surgery—therefore, appropriate pain management is a critical aspect of surgical care [6]. Undertreated postoperative pain has been associated with several short- and long-term disabilities, such as protracted hospitalization, long-term cognitive impairment, and chronic pain syndrome [7,8,9]. Such sequelae have far-reaching socio-economic implications and may subsequently place further downstream stress on the global healthcare system [10]. At the same time, pain management has been marred by over-medication and excessive opioid administration, significantly contributing to the ongoing opioid crisis that has devolved into a global public health crisis [11]. Consequently, it is the clinician’s duty to find a balance in managing postoperative hyperalgesia with empathy, responsibility, and adherence to evidence-based guidelines [12]. 

Although acute postoperative pain (APP) in major abdominal surgery is managed with a variety of drugs, e.g., systemic non-steroidal anti-inflammatory drugs, acetaminophen, ketamine, lidocaine, cyclooxygenase-2 inhibitors, α2-agonists, and/or epidural local anesthetics, opioids remain a cornerstone of every analgesia protocol. In addition to their addictive profile, opioids have a slew of side effects, such as respiratory depression, decreased gastrointestinal motility, and postoperative nausea and vomiting (PONV) [13,14]. To address these issues, opioid-sparing or opioid-free analgesia that relies on non-opioid multimodal analgesic agents is being developed to reduce or completely eliminate opioid use in this setting [15]. Preliminary findings have been reporting promising results, as such approaches can achieve consistent pain relief with a pronounced opioid-sparing effect [16]. Although epidural analgesia (EA) represents, nowadays, the standard of care in specialized centers for major abdominal surgery, the impact on postoperative complications such as PONV and POCD are still debatable [17,18] 

The objectives of this study were to analyze if opioid-sparing analgesic strategies such as epidural analgesia (EA) and a ketamine–lidocaine co-analgetic regimen could impact postoperative pain, as the primary outcome, and to compare the intraoperative fentanyl dose, total postoperative morphine dose, intraoperative sevoflurane concentration, the incidence of POCD, and the severity of PONV, as quantified by postoperative ondansetron dose in the above-mentioned analgetic strategies, as secondary outcomes. 

## 2. Materials and Methods

### 2.1. Study Design. Inclusion/Exclusion Criteria

The study was conducted in accordance with the Declaration of Helsinki and approved by the Local Ethics Committee (protocol code 8999/10 February 2021) for studies involving humans. Informed consent was obtained from all subjects included in the study.

This is a comparative, monocentric, observational, randomized clinical study including patients proposed for major abdominal surgery. We used a convenience sample size of the data. We excluded the patients with inherited or acquired coagulopathies, local inflammation/infection at the epidural punction site, major arrhythmias, heart failure with reduced ejection fraction, chronic pain, and severe dementia.

The patients received balanced anesthesia consisting of inhalative sevoflurane (S.C. Rompharm Company S.R.L, Otopeni, Romania) and one of the following intravenous analgesia regimens: (1) Standard analgesia consisting of intraoperative fentanyl (Chiesi Pharmaceuticals GmbH, Vienna, Austria) (2 mg/kg at induction, followed by repeated doses, as needed) and postoperative acetaminophen (S.C. Santa S.A., Brasov, Romania) (3 g/day) and metamizole (S.C. Zentiva S.A., Bucharest, Romania) (4 g/day); (2) ketamine–lidocaine co-analgesia consisting of lidocaine 1% (S.C. Zentiva S.A., Bucharest, Romania) 1 mg/kg at induction followed by 1 mg/kg/h during surgery, and ketamine (Panpharma, La Selle-en-Luitré, France) 0.5 mg/kg at induction followed by 0.5 mg/kg/h during surgery additionally to standard intraoperative analgesia regimen. Postoperatively, the patients received ketamine 0.5 mg/kg/h, acetaminophen (3 g/day), metamizole (4 g/day), and (3) EA with ropivacaine 0.5% (S.C. Fresenius Kabi Romania S.R.L, Brasov, Romania) (10–14 mL initial bolus considering various patient and surgery-related features), followed by 8–10 mL/h ropivacaine 0.5% intraoperatively additionally to standard intraoperative analgesia regimen. Postoperatively, the patients received EA with 0.2 mg/kg/h ropivacaine 0.2% and fentanyl 0.1 microg/kg/h additionally to acetaminophen (3 g/day) and metamizole (4 g/day). 

All patients received morphine (Lannacher Heilmittel G.m.b.H, Lannach, Austria) as rescue therapy for pain management after surgery.

Postoperatively, we used the visual analogue scale (VAS) as the primary outcome to quantify pain, both at rest and movement. 

We considered the following secondary outcomes: Total intraoperative fentanyl dose, total dose of morphine, intraoperative end-tidal sevoflurane concentration, confusion assessment method for intensive care units (CAM-ICU) score for POCD at 1 h, 6 h, and 12 h postoperatively, and the total dose of ondansetron as a marker for the intensity of PONV. We considered the POCD present in any patient who exhibited positive CAM-ICU test at least once during the testing time.

### 2.2. Data Collection and Analysis

The responsibility of data collection for each patient was assumed by the case-designated anesthesiologist one day prior to surgery. 

The VAS score at 1 h, 6 h, and 12 h after surgery, as the primary outcome, was documented by the ICU nurse, in accordance with the patients’ response. In the intraoperative setting, the designated anesthesiologist considered additional analgesic doses of fentanyl mandatory if any indirect signs of pain appeared, such as tachycardia, hypertension, sweating, reactive mydriasis, patient movement during surgery, and/or unsynchronized breathing patterns. Morphine was administrated postoperatively by the ICU nurse, titrated at a dose of 0.1 mg/kg, in accordance with the local protocol by a VAS score greater than 6 points at rest. 

The target of the intraoperative end-tidal sevoflurane concentration was at the discretion of the designated anesthesiologist in accordance with the above-mentioned patients’ indirect signs of pain or inadequate anesthesia depth and documented accordantly. 

The CAM-ICU scoring was assessed by the ICU nurse at the time points considered, based on a specific flowsheet, and quantified by the designated anesthesiologist on the second day. Ondansetron was administrated by the ICU nurse if any sign of nausea and vomiting occurred, not exceeding 32 mg/day, in accordance with the local protocol. 

Statistical analysis of the data and graphical representations were performed using GraphPad Prism 6.00 (GraphPad Software Inc., California, USA). We evaluated the groups for normal distribution using the D’Agostino–Pearson omnibus normality test and the Shapiro–Wilk test. For comparative analysis, we considered VAS at 1 h, 6 h, and 12 h after surgery and, fentanyl dose (µg/kg/h), morphine dose (mg/kg), sevoflurane (% in expired air), POCD (binary data), and ondansetron dose (mg/day) as markers for PONV severity.

The VAS scores at 1 h, 6 h, and 12 h were considered discrete quantitative data and were expressed as the median and IQR. The medians of the groups were compared using the Kruskal–Wallis test. 

For quantitative data, we used a one-way ANOVA followed by the Bonferroni correction technique for interactions between groups. The fentanyl, morphine, sevoflurane, and ondansetron doses were expressed as quantitative continuous variables and were compared as the mean ± SEM. A two-sided *p*-value < 0.05 was considered statistically significant. For categorical data (i.e., POCD), we used the Chi-squared test.

## 3. Results

### 3.1. Demographic Data

We enrolled 109 patients between February 2021 and January 2022. Of the enrolled patients, 29 refused to participate and 80 patients were assigned to one of the three treatment branches using an urn randomization technique: Standard analgesia (C group, *n* = 32), co-analgesia (Co-A group, *n =* 26), and EA (EA group, *n =* 22). A further 19 patients were excluded from the statistical analysis because they encountered complications during surgery and remained intubated for more than 24 h, required early reintubation, or the epidural catheter was dislocated. The data are displayed in Figure 1. 

Demographic data, such as social status, gender, age, oncologic surgery, and neurodegenerative pathology, are summarized in Table 1. Neurological conditions refer to previous cerebral insults.

### 3.2. The Impact of Multimodal Analgesia Strategies on VAS Score

Pain intensity 1 h after surgery was significantly decreased in Co-A and EA groups, when compared to the C group: C-8 (IQR 6–8) vs. Co-A-6 (IQR 4–7) (*p* = 0.047) and C-8 (IQR 6–8) vs. EA-3 (IQR 2–5) (*p* < 0.0001). Moreover, the pain intensity was even lower in the EA group when compared to the Co-A group: EA-3 (IQR 2–5) vs. Co-A-6 (IQR 4–7) (*p* = 0.036) The same tendency was also observed at 6 and 12 h. At 6 h, the following VAS scores were recorded: C-8 (IQR 7–9) vs. Co-A-4 (IQR 2–5) (*p* = 0.0003), and C-8 (IQR 7–9) vs. EA-2 (IQR 2–4) (*p* < 0.0001). At 12 h, the following VAS scores were observed: C-5 (IQR 3–7) vs. Co-A-3 (IQR 2–4) (*p* = 0.02), and C-5 (IQR 3–7) vs. EA-2 (IQR 2–4) (*p* = 0.001). (Figure 2A–C).

### 3.3. The Impact of Multimodal Analgesia Strategies on the Secondary Outcomes: The Total Intraoperative Fentanyl Dose, Total Postoperative Morphine Dose, Maximal Intraoperative Sevoflurane Concentration, the Incidence of POCD, and the Postoperative Dose of Ondansetron

There was a statistically significant decrease in the mean dose of intraoperative fentanyl used in the Co-A and EA groups, as compared to the C group: C vs. Co-A (4.69 ± 0.76 vs. 1.96 ± 0.16, *p* = 0.0027), C vs. EA (4.69 ± 0.76 vs. 1.18 ± 0.39, *p* = 0.0005). No statistically significant difference was observed between Co-A and EA groups (Figure 3A).

The mean postoperative morphine dose per kg revealed a statistically significant decrease in Co-A and EA groups when compared to the C group: C vs. Co-A (0.16 ± 0.02 vs. 0.09 ± 0.016, *p* = 0.037), C vs. EA (0.16 ± 0.02 vs. 0.04 ± 0.013, *p* = 0.0005) (Figure 3B).

Comparing the means of sevoflurane concentration in exhaled air, the data revealed a statistically significant decrease in the EA group compared to the C group (1.47 ± 0.05 vs. 1.63 ± 0.04, *p* = 0.042) and between the Co-A and EA groups (1.63 ± 0.18 vs. 1.47 ± 0.05, *p* = 0.045) (Figure 3C).

The highest incidence of POCD was observed in the C group (24%), followed by the Co-A group (16%) and the EA group (8%). The difference was, however, statistically significant between the C and EA groups only (*p* = 0.0085) (Figure 4A). The comparative analysis of ondansetron dose postoperatively revealed no statistically significant differences among groups: C vs. Co-A (1.38 ± 0.54 vs. 0.73 ± 0.3, *p* = NS), C vs. EA (1.38 ± 0.54 vs. 0.74 ± 0.31, *p* = NS) (Figure 4B).

## 4. Discussion

All major surgical procedures are associated with APP, which, if not managed properly, can not only increase comorbidity burden but also progress to chronic postsurgical pain [19]. Furthermore, inadequate pain management in elderly patients who underwent major abdominal surgery could lead to cognitive impairment, insomnia, longer ICU stay, and delays in postoperative recovery [20]. As such, anesthesiologists are uniquely positioned to influence perioperative opioid use and misuse in APP management through the incorporation of opioid-sparing individual multimodal analgesic agents into modified analgesia protocols. Considering that nociception monitoring during surgery is less reliable and not included in the standard of care, the achievement of this goal is even more challenging [21].

Currently, conclusions are not yet robust enough to definitively alter prescription practices, as studies examining APP management via alternative opioid-sparing or opioid-free analgesia regimens continue to report results that are heterogeneous or lack scope [22]. In this study, we added to the body of growing evidence supporting opioid-sparing analgesia in major abdominal surgery by testing two modified regimens, which considerably reduced perioperative analgesia requirements, with either systemic ketamine combined with lidocaine, or with epidural analgesia.

The effect estimates of pain at early time points in this study (1–6–12 h) indicate that patients undergoing major elective abdominal surgery who received an individualized multimodal analgesic strategy, either through epidural analgesia or co-analgesics, experienced less pain than patients receiving the standard analgesia protocol. The group receiving the intraoperative systemic ketamine–lidocaine analgesic regimen, followed by systemic postoperative lidocaine, reported less pain, coupled with a reduction in cumulative intra- and postoperative opioid consumption. We consider such a reduction to be clinically significant—in select patient populations that require higher dosages of opioids (e.g., cancer or drug-dependent patients), for adequate pain control, alternative analgesia regimens may be particularly beneficial.

In a systematic review, Bell et al. found that 27 of 37 studies included in the meta-analysis reported clear benefits of a perioperative subanesthetic dose of ketamine administration and a positive correlation with a reduction in rescue analgesia requirements and/or pain intensity [23]. However, the authors concluded that no specific administration guidelines could be extracted due to striking heterogeneity among the reported studies. Further randomized controlled trials are required to address the lack of consensus regarding the dosing strategy, therapy duration, and modality of administration. Small studies may overestimate the treatment effect and overlook the occurrence of adverse events. Furthermore, it is also unclear if ketamine prevents the development of residual pain or mechanical hyperalgesia, as rigorous long-term follow-up across various surgery types has not been performed.

Specifically for major abdominal surgery, systemic ketamine administration has been evaluated in several studies, both as a single intravenous bolus and continuous systemic infusion [24,25,26]. Doses for bolus injections were 0.5 mg/kg, while dosages for systemic infusions varied from 0.12 mg/kg/h to 0.25 mg/kg/h. Both studies reported decreased postoperative pain scores and opioid needs. Kock et al. further reported diminished residual pain at a six-month follow-up [25].

However, even when strictly within the scope of abdominal surgery, the parameters of drug delivery are ambiguous. Katz et al. compared preincisional ketamine to continuous infusions in male patients undergoing radical prostatectomy and found no differences in pain incidence or intensity [26]. Ketamine could also selectively target the affective component of pain, providing a durable decrease in postoperative pain that outlasts its known half-life, but further research is required to confirm this finding [27].

Likewise, systemic lidocaine use as a multimodal analgesic for postoperative pain is flanked by compelling evidence. A review of 2802 patients across 45 studies found that the use of intravenous lidocaine infusion reduced postoperative pain at early and intermediate time points, while also affording faster gastrointestinal recovery, as compared to epidural anesthesia with opioids [28]. In laparoscopic abdominal surgery, lidocaine infusions reduced cumulative postoperative morphine consumption by 50–66% while improving pain scores within the first 48 h after surgery [29,30]. In bariatric surgery, lidocaine also improved recovery scores and reduced opioid consumption [31,32,33,34].

The synergistic effect of lidocaine and ketamine in clinical trials on pain management in abdominal surgery is still debatable [35]. Our results revealed a potential benefit of this analgesic strategy to pain intensity after major abdominal surgery. Consistent with our findings, such combined treatment proves its superiority in experimental settings [36,37].

In patients who received epidural analgesia, we observed that the intensity of the pain at 1 h after surgery was reduced even further, as compared to the ketamine–lidocaine regimen. At 6 and 12 h after surgery, the pain scores were similar. Moreover, perioperative systemic opioid doses were also reduced compared to control and were comparable between the two multimodal regimens. Our results are in accordance with El Sayed et al., who found that epidural analgesia in major abdominal surgery improves pain control [38]. Thus, we demonstrated the viability of opioid-sparing strategies to decrease postoperative pain intensity, as the primary outcome, and reduce perioperative opioid consumption (fentanyl and morphine) as the secondary outcome.

Considering the intraoperative hypnosis intensity in patients undergoing major abdominal surgery, as one of the secondary outcomes, we observed a reduction in sevoflurane concentration secondary to EA. Consistent with our findings, Panousis et al. reported reduced intraoperative hypnotic use. Moreover, further benefits such as better intraoperative fluid management and a reduction in the catecholamines requirement could contribute to the improvement of the anesthesia quality [39].

Our data do not support the reduction in the intraoperative sevoflurane concentration when combined with ketamine/lidocaine. Conversely, some experimental studies identified the combined administration of ketamine and lidocaine as synergistic, where the minimum alveolar concentration of volatile anesthetics was reduced by up to 57% without affecting anesthesia quality or recovery times [40,41]. Other studies identified a potential benefit of lidocaine [42]. Further clinical studies are needed to confirm this experimental paradigm.

Efforts should be made to reduce the incidence of POCD, especially in the elderly, as it could deteriorate the patients’ ability of social reintegration. Our study revealed a reduction in POCD incidence in patients receiving EA. The results are in accordance with Orhun et al., who concluded that the reduction of the POCD incidence could be secondary to better pain control [18]. The exact mechanism is, however, still unknown. Furthermore, in our studies, the patients receiving perioperative ketamine and lidocaine did not experience the same reduction in POCD incidence, despite the fact that they experienced less pain.

Finally, we observed no difference in PONV severity among the groups, as revealed by ondansetron dose. Xie et al. reported that lidocaine infusion could accelerate recovery after surgery in patients with obstructive sleep apnea, in part due to a reduction in PONV incidence [34]. Considering the impact of EA on the PONV severity, the data in the literature are heterogeneous [43]. A more comprehensive analysis, including patients or surgical procedures at risk for PONV, should further address this issue [44].

### Study Limitation

The primary outcome of our study was the VAS score. Consequently, the patients who developed postoperative respiratory failure, needed reintubation, experienced surgery-related complications, or needed reintervention were excluded because of their reduced ability to answer the investigator questions or the exceedance of the 24 h observational time.

Although postoperative monitoring was performed by the case anesthesiologist, no double blinding protocol was used. The role of perioperative nociception monitoring with commercially available systems and/or integrated nursing information systems is still debatable. Thus, our study design did not include such pain management strategies, focusing on perioperative pain therapy concepts only.

The low number of patients included in the study is based on the assumed statistical power of over 80% for the primary endpoint (VAS score at 1 h, 6 h, and 12 h). Even if no further assumptions were made on the secondary outcomes, our results raise awareness about this original and controversial topic: Opioid-sparing anesthesia in major abdominal surgery.

## 5. Conclusions

Multimodal analgesia concepts should be used in accordance with patient consent and needs. Local protocols should be issued to find an individualized approach of APP management, which considers surgery indication, the type of surgery required, and the potential contraindications for an invasive pain management strategy. Efforts should be made to reduce opioid consumption in a perioperative setting. Further clinical studies are required to improve various opioid-free or opioid-sparing pain management strategies.

## Figures and Tables

**Figure 1 medicina-58-00487-f001:**
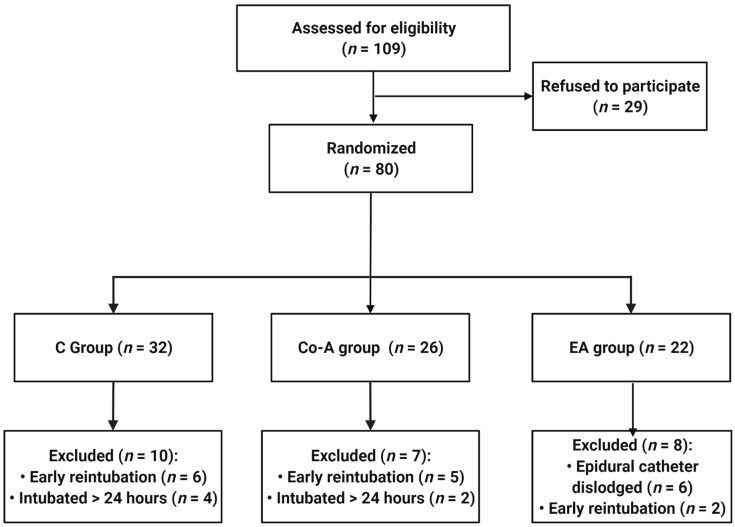
Flow chart representing patient selection for the study.

**Figure 2 medicina-58-00487-f002:**
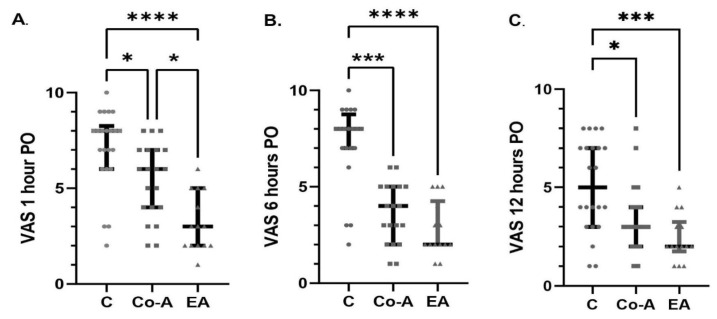
The Visual Analogue Scale (VAS) scores at 1 h (**A**), 6 h (**B**), and 12 h (**C**) postoperatively (PO) among groups. Horizontal and vertical lines represent the medians and the IQR of each group, respectively. * *p* = 0.05–0.01, *** *p* = 0.001–0.0001, and **** *p* < 0.0001.

**Figure 3 medicina-58-00487-f003:**
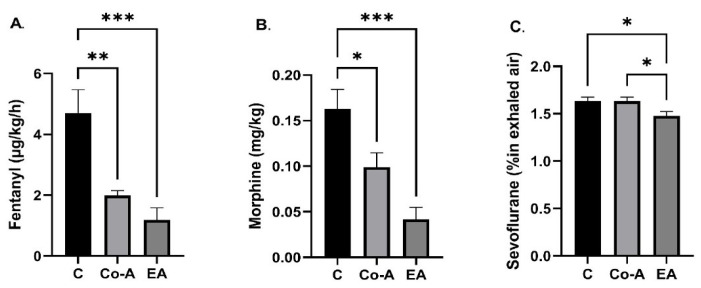
The mean intraoperative fentanyl (µg/kg/h) (**A**), the postoperative morphine (mg/h) (**B**), and the intraoperative sevoflurane doses (%) (**C**) among groups. Bars are mean ± SEM. * *p* = 0.05–0.01, ** *p* = 0.01–0.001, and *** *p* = 0.001–0.0001.

**Figure 4 medicina-58-00487-f004:**
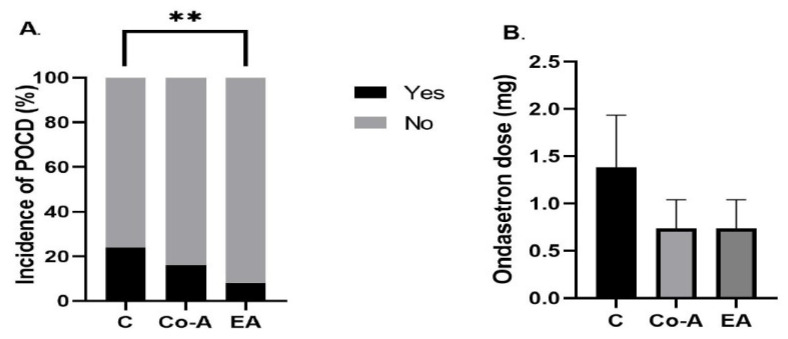
The incidence of (**A**) POCD in % and (**B**) the mean postoperative dose of ondansetron (mg) among groups. Bars are mean ± SEM. ** *p* = 0.01–0.001.

**Table 1 medicina-58-00487-t001:** Demographic data among study groups.

	Control (*n =* 22)	Co-A (*n =* 19)	EA (*n* = 14)
Poor social status (%)	36.36	40.76	60
Female gender (%)	40.9	47.39	76.92
Mean age (years)	52.27	61.2	69.53
Oncologic surgery (%)	72.72	63.15	92.3
Neurological condition (%)	22.72	5.26	15.3

## Data Availability

Not applicable.

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
