# Peer review of "Opioid-Sparing Analgesia Impacts the Perioperative Anesthetic Management in Major Abdominal Surgery"

_medicina, 2022, doi:10.3390/medicina58040487_

Round 1

Reviewer 1 Report

Thanks for sharing this manuscript for review.

The objective of the study lacks originality. The analgesic epidural is currently a standard of practice that does not require demonstrating its effectiveness.

The study is adequately executed, but the manuscript requires an arrangement of paragraphs and consistency between the different sections to give credit to the effort put into the study’s realization. Also, the absence of double-blinding limits the interpretation of the results.

Abstract:

- Please subdivide the outcomes into primary outcome and secondary outcomes

- Please sharply state the study objective and the main research hypothesis

- The results must follow the same description of the primary endpoint first and then the secondary ones.

Main manuscript

- Line 78 and 79: Please clearly distinguish the primary and secondarily, the other outcomes and describe the study hypothesis.

- Line 85: Please, specify the ethics committee's acceptance and the patient's agreement to participate

- Line 85: Please specify the randomization method

- The inclusion and exclusion criteria should come before the description of the groups

- The number of patients included, the periods of inclusion, and the demographic data should be in the results section

- The collected data should be organized as follows: Please start with the primary endpoint, then the secondary ones. Explain how the data are collected (by whom, what scale, etc.).

- Statistical analysis should follow the same path: For each data, please describe how it will be presented in the text and the statistical test used to compare it between the groups, keeping the same order of the primary outcome first, then the secondary ones.

- Discussion should also follow the same path.

- I would not consider the Visual Analogue Scale (VAS) score as continuous data expressed in mean (SD). It is discontinuous data expressed in Median [IQR].

A study limitation section should be added

- No double blinding

- No postoperative respiratory complications or surgical complications were recorded, as these are complications potentially related to postoperative morphine consumption.

- Was there a sample size calculation? Was it performed according to the primary endpoint? Was it a convenience sample? Thank you for specifying.

Author Response

The objective of the study lacks originality. The analgesic epidural is currently a standard of practice that does not require demonstrating its effectiveness.

We thank the reviewer for the opportunity to address this issue. Although, EA represents, nowadays, standard of care in specialized centers for major abdominal surgery, the impact on postoperative complications like PONV and POCD are still debatable. We added this information in the introduction and documented reference (https://pubmed.ncbi.nlm.nih.gov/29936600/, https://pubmed.ncbi.nlm.nih.gov/31942729/). Moreover, other analgesia strategies like lidocaine-ketamine as alternative to epidural anesthesia, represents, nowadays, an innovative and controversial issue.

The study is adequately executed, but the manuscript requires an arrangement of paragraphs and consistency between the different sections to give credit to the effort put into the study’s realization. Also, the absence of double-blinding limits the interpretation of the results.

Abstract:

  1. Please subdivide the outcomes into primary outcome and secondary outcomes

We stated the primary and secondary outcomes accordantly (line 35-40)

  1. Please sharply state the study objective and the main research hypothesis

We added more concise data regarding the objectives of our study (line 27-28)

  1. The results must follow the same description of the primary endpoint first and then the secondary ones.

 We illustrated the results in accordance with the primary and secondary outcomes (line 40-44)

Main manuscript

  1. Line 78 and 79: Please clearly distinguish the primary and secondarily, the other outcomes and describe the study hypothesis.

We thank the reviewer for this valuable comment. We reconsidered the paragraph accordantly (line 82-87)

  1. Line 85: Please, specify the ethics committee's acceptance and the patient's agreement to participate

We retained form the Journal format that a separate section (at the end of the manuscript, before Bibliography), should contain this information. We added the sentences supplementary also before M and M as requested (lines 90-92). If any other issues should be addressed here, please specify.

  1. Line 85: Please specify the randomization method technic

We thank the reviewer for addressing this issue. We used an urn randomization technic (considering 2 treatments and control). The information was added at the lines 162-163, in the results section-demographic data

  1. The inclusion and exclusion criteria should come before the description of the groups

We reconsidered the paragraphs (lines 93-96)

  1. The number of patients included, the periods of inclusion, and the demographic data should be in the results section

We moved these informations as required (lines 159-165).

  1. The collected data should be organized as follows: Please start with the primary endpoint, then the secondary ones. Explain how the data are collected (by whom, what scale, etc.).

We thank the reviewer for this important comment. We added the information requested in the subsection 2.3 entitled now “Data collection and analysis” (lines 126-142)

  1. Statistical analysis should follow the same path: For each data, please describe how it will be presented in the text and the statistical test used to compare it between the groups, keeping the same order of the primary outcome first, then the secondary ones.

We thank the reviewer for the valuable comment. We would like to mention that we rewrote the paragraph about data analysis, regrouping the variables by primary outcome first (VAS score), and secondary ones, considering that a subgrouping of secondary outcomes with regard to the statistical test used is also logic and important (lines 143-156). The results section was reorganized into 3 subsections as well: demographic data, VAS score (as primary outcome) and the other secondary outcomes

  1. Discussion should also follow the same path.

We reorganized, as required, the Discussion subsection, considering the VAS score, Fentanyl and Morphine doses as having the same clinical relevance. Moreover, we added some new discussion perspectives considering the other secondary outcomes like: sevoflurane concentration, POCD and PONV. Two other references were added too. Lines 277-281, 288-298, 304-317

  1. I would not consider the Visual Analogue Scale (VAS) score as continuous data expressed in mean (SD). It is discontinuous data expressed in Median [IQR].

We reconsidered the expression of VAS as requested (lines 175-183). Moreover, we designed the Fig.1 in accordance with this new change for more clarity and consistence of the data presented among with the fig legend (scatter plot is more adequate in this case). Additionally, Kruskal-Wallis test were used for comparing the medians among groups, as being more reliable (in accordance with the literature). Although, the statistical significance was the same.   

  1. A study limitation section should be added

- No double blinding

- No postoperative respiratory complications or surgical complications were recorded, as these are complications potentially related to postoperative morphine consumption.

We thank the reviewer for the valuable remark. A study limitation section was added (lines 318-329).

- Was there a sample size calculation? Was it performed according to the primary endpoint? Was it a convenience sample? Thank you for specifying.

We are very thankful for this consideration. We added specific information on this issue in study limitation subsection. Although low, the patients number included in this study followed an initially statistical power calculation focused on the main outcome: VAS. We considered a statistical power over 80% to be adequate, in accordance with the literature. We stated further, that the supplementary discussion on the results of the secondary outcomes considered, should rise the clinician awareness on this original topic, even if the statistical power/sample size calculation focused on the main outcome only. (lines 318-329)

Reviewer 2 Report

The references of the introduction are national or local  (2-5). The number of patients included in each group is low. No flow chart diagram. No statiscatical information of the number of patients needed for the objetives. They do not use any monitor of nociception. They do not drescribe in anaesthetic technique the use of sevoflurane.

Author Response

Reviewer 2

  1. The references of the introduction are national or local (2-5).

These references are national studies conducted in two reference centers for major abdominal surgery

  1. The number of patients included in each group is low.

We thank the reviewer for addressing this issue. We recognize the limitation of a monocentric, observational study, which per definition could include, statistically, a variable number of samples. We added, accordantly, a subsection entitled study limitation, focusing on statistical power with regard to the main outcome considered. Additionally, we stated the statistical impact of the study design on the data interpretation, considering the secondary outcomes, as reliable for raising the clinician awareness on this original and debatable topic: opioid-sparing anesthesia and analgesia in abdominal surgery (lines 318-329).

  1. No flow chart diagram.

We thank the reviewer for this constructive comment. We added a flowchart diagram in the results section as Fig.1, as requested.

  1. No statiscatical information of the number of patients needed for the objetives.

We thank the reviewer for this valuable comment. We added in a new subsection, study limitation, an explanation focused on the statistical power calculation and sample size (lines 318-329).

  1. They do not use any monitor of nociception.

We are thankful for this consideration. We added consequently in the discussion section the actual state of knowledge in this area. Because the nociception monitoring is not included, nowadays, in the standard of care, we considered the indirect signs to be equally valuable (lines 226-228). Please consider the supplementary reference (No.21), which support our statement.

  1. They do not drescribe in anaesthetic technique the use of sevoflurane.

We added this information accordantly. (Line 97)

Round 2

Reviewer 1 Report

I thank the authors for ensuring the suggested modifications.

The quality of the manuscript is now improved.

Here are other essential suggestions as a result of the proofreading of this new version:

- Line 93: The number of patients should appear at the beginning of the results section

- Other limitations: There is no NIS or NOL monitoring. It would be very relevant for such a study comparing intraoperative nociception and sevoflurane consumption.

  • It is not usual to find that the groups are assigned after exclusions. Following the patient's consent, he is usually included in a group. I also noted that patients who received epidural analgesia were excluded. However, if they received epidural analgesia, they should be in the EA group. The flowchart should be thoughtfully reviewed and not confusing.

- In table 1: Why did you compare the social status of the patients? How does this interfere with postoperative analgesia? It would be wiser to compare lifestyle habits such as previous consumption of narcotics or other painkillers. Also, please explain “Neurological conditions.”

- Statistics:

. The primary endpoint should ideally be the EVA at a specific time, such as 1h, and the other comparisons (6h and 12h) should be secondary endpoints. Although, as it is done, it is acceptable. I leave the decision to the editor's discretion.

. The P-value should be corrected within the component of the primary outcome and the secondary outcomes using the Bonferroni technique (multiple endpoints).

. Limitations: Was the sample size previously estimated? Please specify. Otherwise, please specify if a convenience sample size was used.

I hope this will help.

Issam.

Author Response

I thank the reviewer for the new comments.

Further, please find the point by point answers to your comments 

  1. Line 93: The number of patients should appear at the beginning of the results section.

We thank the reviewer for addressing this issue. We erased the number of the patients at line 93 and added at the beginning of the results section, as requested.

-2. Other limitations: There is no NIS or NOL monitoring. It would be very relevant for such a study comparing intraoperative nociception and sevoflurane consumption.

We are thankful the reviewer for considering this issue. We addressed this topic in the discussion section, mentioning that the quantification of nociception during surgery is difficult and the role of the anesthesiologist even more challenging in pain treatment, especially in major abdominal surgery. According to Ledowski et al (ref.21 form the manuscript), the available commercial solutions for assessing nociception during surgery are not superior and it’s not representing, nowadays, standard of care. Thus, we evaluated the intraoperative nociception using standard vital parameters as described at lines 129-133.

Additionally, a NIS is usually (not necessary but major publications are focusing on this topic) designed for cancer patients, being widely used for assessing pain intensity variation in time, as response to therapy (doi: 10.1371/journal.pone.0222516). These approach, although complex and reliable, is different form our study design, who evaluates absolute values of VAS at different time point and compared among groups. Moreover, NIS, a promising tool in pain medicine, it is remain the solution for the future, with a very promising research potential per se. Our study focused on perioperative therapy concepts based on the standard of care outcome variables.

We added, however, as requested, this issue in the study limitation section. (lines 326-329)  

  1. It is not usual to find that the groups are assigned after exclusions. Following the patient's consent, he is usually included in a group. I also noted that patients who received epidural analgesia were excluded. However, if they received epidural analgesia, they should be in the EA group. The flowchart should be thoughtfully reviewed and not confusing.

Many thanks for highlighting this topic. We designed a new flowchart as requested, in which we included the 25 patients, who experienced complications that could interfere with the quantification of the outcomes in the study groups (19 related to reintubation or longer intubation and 6 related with epidural dislodgement). For more clarity, after group inclusions were made, we illustrated the patients excluded from each group, as a result of either one of the complications mentioned: early reintubation, intubated longer than 24h, catheter dislodgement. Please consider appropriate to change the numbers of the patients included in each study group accordantly: please reconsider the lines 29-33 and 159-165.

  1. In table 1: Why did you compare the social status of the patients? How does this interfere with postoperative analgesia? It would be wiser to compare lifestyle habits such as previous consumption of narcotics or other painkillers. Also, please explain “Neurological conditions.”

     We thank the reviewer for considering this issue.

     Firstly, the social status is usually linked with the patients perceived distress magnitude secondary to stress situations like surgery. It is well-known that an optimal stress managing involves an appropriate educational and understanding background. That’s way, patients with a poor socioeconomic status could be more distressed to the same stressor condition (i.e. surgery). Furthermore, there are sufficient data in the literature, who sustained that perioperative complications like pain, cognitive dysfunction and PONV are correlated with the estimated patient’s distress level. We added this condition in our study, in order to eliminate this confounding factor among groups.

     Secondary, the study design focused on three pain therapy concepts in acute settings. No chronic pain patients were included, being considered confounding factor, as well. We added accordantly, this information in the subsection referring exclusion criteria (line 96).

     Finally, neurological conditions refer to previous vascular cerebral insults episodes which could impair the postoperative moving ability and cognition. All our patients with neurological diseases included in the study, had, however, no or minor motoric impairments, that did not interfere with the outcomes considered. Additionally, we added this information (lines 172-173)      

- Statistics:

  1. The primary endpoint should ideally be the EVA at a specific time, such as 1h, and the other comparisons (6h and 12h) should be secondary endpoints. Although, as it is done, it is acceptable. I leave the decision to the editor's discretion.

Thank you very much for your comment.

  1. The P-value should be corrected within the component of the primary outcome and the secondary outcomes using the Bonferroni technique (multiple endpoints).

Thank you for this useful comment. We revised, as requested, the p-values using Bonferroni correction technique. The p-values were corrected. (lines 152-153)

  1. Limitations: Was the sample size previously estimated? Please specify. Otherwise, please specify if a convenience sample size was used.

We are thankful for this consideration. We used a convenience sample size. The information was added at line 93-94. After statistical analysis, the sufficiency of our sample size was confirmed by the statistical power of over 80% (data added, as requested, in the previous reviewing stage, in study limitation section at lines 330-331) for the primary outcomes considered.

Best Regards,

Sebastian

Reviewer 2 Report

new version Ok

Author Response

Thank you very much for the reviewing process